

# Variability and Predictability of a reduced-order land atmosphere coupled model

Anupama K Xavier[1,2], Jonathan Demaeyer[1], and Stéphane Vannitsem[1]

[1]Royal Meterological Institute of Belgium,Avenue Circulaire 3, 1180, Brussels, Belgium

[2]Université libre de Bruxelles, de la Plaine 155, 1050, Brussels, Belgium

**Correspondence:** Anupama K Xavier (xavier.anupamak@meteo.be)

**Abstract.** This study delves into the predictability of atmospheric blocking, zonal, and transition patterns utilizing a simplified coupled model. This model, implemented in Python, emulates midlatitude atmospheric dynamics with a two-layer quasi-geostrophic channel atmosphere on a beta-plane, encompassing simplified land effects. Initially, we comprehensively scrutinize the model's responses to environmental parameters like solar radiation, surface friction, and atmosphere-ground heat exchange.

Our findings confirm that the model faithfully replicates real-world Earth-like flow regimes, establishing a robust foundation for further analysis. Subsequently, employing Gaussian mixture clustering, we successfully delineate distinct blocking, zonal, and transition flow regimes, unveiling their dependencies on surface friction. To gauge predictability and persistence, we compute the averaged local Lyapunov exponents for each regime. Our investigation uncovers the presence of zonal, blocking, and transition regimes, particularly under conditions of reduced surface friction. As surface friction increases further, the system

transitions to a state characterized by two blocking regimes and a transition regime. Intriguingly, periodic behavior emerges under specific surface friction values, returning to patterns observed under low friction coefficients. Model resolution increase impacts the system in a way that only two regimes are then obtained with the clustering: the transition phase disappears and the predictability drops to roughly 2 days for both of the remaining regimes. In accordance with previous research findings, our study underscores that when all three regimes coexist, zonal patterns exhibit a more extended predictability horizon compared

to blocking patterns. Remarkably, transition patterns exhibit reduced predictability when coexisting with the other regimes. In addition, within a specified range of surface friction values where two blocking regimes are found, it is observed that blocked atmospheric situations in the west of the applied topography are marked by instabilities and reduced predictability in contrast to the blockings appearing on the eastern side of the topography.



# 1  Introduction and motivation

Low-Frequency Variability (LFV) encompasses a wide range of atmospheric and climate processes, including atmospheric blockings, heat waves, cold spells, and long-term oscillations like the Madden-Julian Oscillation (MJO), the North Atlantic OScillation (NAO), and the El-Ni/ño - Southern Oscillation (ENSO). Despite extensive research, a comprehensive understanding of the nature of these LFVs remains elusive. In practical terms, exploiting these LFVs to achieving accurate extended-range forecasts beyond two weeks at midlatitudes remains a formidable challenge. On the climate front, comprehending how climate
change affects the low-frequency variability of the atmosphere also remains an area of incomplete knowledge. Previous works, as reported in Ghil and Robertson (2002), Lucarini and Gritsun (2020), have highlighted this gap in understanding.

Blocking systems - a notable form of LFV observed in the atmosphere - can be described as long-lasting, quasi-stationary flow patterns in the troposphere (Liu, 1994). These patterns are characterized by a significant meridional flow component, leading to a disruption or deceleration of the zonal westerly flow at midlatitudes (Nakamura and Huang, 2018). While the
blocking systems persist, strong zonal flows may simultaneously exist to the north and south of them. The evolution of blocking systems involves transitions between more zonal and more meridional flow patterns during their onset and decay phases, posing challenges for forecast models (Frederiksen et al., 2004). Moreover, the dynamics of blocking systems are complex, involving interactions across different spatial and temporal scales, both internally within the system and with the surrounding flow environment (Shutts, 1983; Lupo and Smith, 1995). Researchers have highlighted the intricate nature of these dynamics
and the connections between various scales, contributing to the challenges in understanding and predicting the behavior of blocking systems. As blocking systems have the potential to induce weather extreme like heatwaves, there is a notable interest in understanding how the characteristics of these blocking events might evolve in the future and how such changes could subsequently impact the occurrence and features of surface extreme weather events. The investigation of these potential changes is of significant importance to assess the risks associated with extreme weather events and to enhance our understanding of
the complex interactions between blocking patterns and surface weather conditions in a changing climate context (Kautz et al., 2022).

Eventhough the concerns above matters, identifying and evaluating LFVs in GCMs is computationally expensive, so in this study an idealised reduced order coupled model is used. It is a climate model 'stripped to the bone', which links theoretical understanding to the complexity of more realistic models, made by key ingredients and approximations; which hence helps
us to study a particular phenomenon by tweaking the parameters affecting them with less computational cost. These types of simplified models age back from Lorenz (1963) where he demonstrated the presence of the property of sensitivity to initial conditions in a simple system, that subsequently led to the development of chaos theory.

Later on the trend continued with Charney and DeVore (1979) in which a quasi-geostrophic model, projected onto Fourier modes for a more efficient and concise representation, has been developed. It also incorporates an idealized parameterization
closure to account for subgrid-scale processes. By imposing in addition a meridional temperature gradient over a topography, the model becomes a forced-dissipative system, which exhibits multiple stable equilibria, representing distinct atmospheric



flow patterns. Charney and Devore hypothesized that the transitions between these solutions were primarily influenced by small-scale perturbations or the presence of baroclinic instability within the system.

Charney and Straus (1980) partially confirmed this hypothesis and found out that these transitions were indeed the result of baroclinic instability. Their study sheds light on the complex interactions between atmospheric flow, orography and propagating planetary waves in baroclinic systems. They discovered that the interactions between atmospheric flow and orography induces form-drag instability, generating eddies and perturbations, and leading to multiple stable equilibria with distinct flow patterns under consistent forcing conditions.

Charney's seminal study sparked significant interest in the low-order spectral model and the theory of multiple flow equilibria. Zhengxin and Baozhen (1982) and Zhu (1985) employed a two-layer low-order spectral model, discovering stable equilibrium states resembling actual blocking, with zonally asymmetric thermal and topographic forcings and flow nonlinearity playing critical roles in blocking dynamics. The summarized version of the evolution of numerical weather prediction and predictability tools are included in Yoden (1983a, b, 2007).

Reinhold and Pierrehumbert (1982, 1985) extended Charney's model, incorporating additional synoptic-scale waves, revealing two distinct weather regime states influenced by wave-wave interactions causing transitions between equilibrium states.

Cehelsky and Tung (1987) demonstrated that the behavior of a reduced-order model exhibits notable disparities at higher resolutions, primarily attributed to the inadequate representation of energy upscaling and vorticity downscaling pathways. They coined this phenomenon as '*spurious chaos*', denoting the emergence of irregular dynamics that are not genuinely representative of the underlying physical processes. Although this is a valid point, high resolution models are usually hard to analyze in detail. There is therefore a need for investigating first reduced coupled model in order to get qualitative conclusions on a problem at hand. We started this journey by using a reduced order land atmospheric model to investigate the impact of coupling between the land and the atmosphere on LFV.

Legras and Ghil (1985) employed a higher-order barotropic spectral spherical model to investigate blocking and zonal flow regimes dynamics, suggesting that their model displayed properties akin to an index cycle, and later stochastic forcing was introduced to Charney's deterministic model, leading to transitions between high- and low-index states (Benzi et al., 1984; Egger, 1981; Sura, 2002). The impact of stochastic forcing on the stability of atmospheric regimes was also recently considered in a highly-truncated barotropic model by Dorrington and Palmer (2023), where they provide a mechanism to explain the increased persistence of blocking due to the noise in such simple models.

Schubert and Lucarini (2016) recent numerical investigation employing a QG model revealed a counter-intuitive finding that during blocking events, the global growth rates of the fastest growing covariant Lyapunov vectors (CLVs) are significantly higher, indicating stronger instability compared to typical zonal conditions. The difficulty in predicting the specific timing of blocking onset and decay further contributes to the observed instability behavior, aligning with Kwasniok (2019) findings associating anomalously high values of finite time largest Lyapunov exponents with blocked atmospheric flows.

Consistent results were obtained by Faranda et al. (2016, 2017), utilizing extreme value theory for dynamical systems, which identified blocking regimes with unstable fixed points in a heavily reduced phase space. Their findings indicated that blockings exhibit higher instability in the circulation, linked to an increased effective dimensionality of the system. This agreement





with Schubert and Lucarini (2016) study further supports the notion that blocking events display stronger turbulence and instability, challenging conventional expectations.

We here aim at extensively investigating the predictability of blocking, zonal, and transition regimes utilizing backward Lyapunov exponents (BLVs) in the context of a recently developed reduced-order land-atmosphere model, providing a more comprehensive understanding of the system's behavior and regime predictability.

The classic Charney's model lacks feedback from atmospheric flow to the artificially specified "thermal forcing," leading to potential unrealistic effects on large-scale atmospheric motions. To address this limitation, a new land atmospheric coupled model is proposed in Li et al. (2018), which incorporate an energy balance scheme to allow atmospheric motions to influence the land temperature distribution and vice-versa. By considering horizontally inhomogeneous radiative input fields as the driving force for land-atmosphere dynamics, this coupled model offers a more realistic representation of the interactions between the land and the atmosphere. The model bears resemblance to the low-order coupled ocean-atmosphere model proposed by Vannitsem et al. (2015), but with a heat bath featuring the land and an idealized topography.

Prior to conducting the investigation on the predictability of blocking, zonal, and transition regimes using backward Lyapunov exponents (BLVs), we performed a characterization of the sensitivity of the quasi-geostrophic land-atmosphere coupled model embedded in the `qgs` framework (Demaeyer et al., 2020) with respect to various environmental parameters that are essential for the functioning of the atmosphere.

The structure of the paper is as follows. Section 2 introduces the model, outlining its structure, main properties, and the parameters employed in the study. Additionally, this section includes a discussion on the stability properties of the system and temporal evolution of the modes (barotropic stream function, baroclinic streamfunction and ground temperature). In Section 3, the methodology used for the investigation is explained. Section 4 presents the stability and the Lyapunov properties of the model corresponding to various environmental factors, and in Section 5, predictability properties of different weather regimes are discussed. Effects of the model resolution are presented in section 6 and the conclusions drawn from the research are provided in section 7, along with future perspectives for further studies.

## 2 Land atmosphere coupled model

### 2.1 Model Characteristics

`qgs` is a Python framework in which several reduced-order climate models are implemented for midlatitudes (Demaeyer et al., 2020). It models the dynamics of a 2-layer quasi-geostrophic (QG) channel atmosphere on a beta-plane, coupled to a simple surface component that could be a land or an ocean. In the current study, we are using the quasi-geostrophic land-atmosphere coupled model version (Li et al., 2018).

The atmospheric part of the model is represented as a 2-layered, quasi-geostrophic flow defined on a $\beta$ plane within the zonal walls $y = 0$ and $\pi L$ (Reinhold and Pierrehumbert, 1982). The thermodynamic equations of the baroclinic atmosphere includes the energy exchanges between land, atmosphere and space similar to the radiative and heat flux scheme provided in Barsugli and Battisti (1998). The coupling of the atmospheric components with the ground is constituted by the surface friction and the



radiative and heat exchanges between the atmosphere and the ground. As usual in such types of models, channel atmosphere is considered with no-flux boundary conditions on the north and south borders and periodic boundary conditions on the east and west border.

The equations governing the time evolution of barotropic and baroclinic streamfunction of the atmospheric part are as follows:

$$\frac{\partial}{\partial t}\left(\nabla^2\psi_\mathrm{a}\right) + J(\psi_\mathrm{a}, \nabla^2\psi_\mathrm{a}) + J(\theta_\mathrm{a}, \nabla^2\theta_\mathrm{a}) + \frac{1}{2}J(\psi_\mathrm{a} - \theta_\mathrm{a}, f_0\,h/H_\mathrm{a}) + \beta\frac{\partial\psi_\mathrm{a}}{\partial x}$$
$$= -\frac{k_d}{2}\nabla^2(\psi_\mathrm{a} - \theta_\mathrm{a}) \tag{1}$$

$$\frac{\partial}{\partial t}\left(\nabla^2\theta_\mathrm{a}\right) + J(\psi_\mathrm{a}, \nabla^2\theta_\mathrm{a}) + J(\theta_\mathrm{a}, \nabla^2\psi_\mathrm{a}) - \frac{1}{2}J(\psi_\mathrm{a} - \theta_\mathrm{a}, f_0\,h/H_\mathrm{a}) + \beta\frac{\partial\theta_\mathrm{a}}{\partial x}$$
$$= -2\,k_d'\nabla^2\theta_\mathrm{a} + \frac{k_d}{2}\nabla^2(\psi_\mathrm{a} - \theta_\mathrm{a}) + \frac{f_0}{\Delta p}\omega \tag{2}$$

where $\omega$ is the verical velocity of the system. $\psi_a$ is the barotropic streamfunction and $\theta_a$ is the baroclinic streamfunction of the atmosphere. The constants $k_d$ and $k_d'$ multiply the surface friction term and the internal friction between layers, respectively. The temperature equation of the baroclinic atmosphere and ground are :

$$\gamma_\mathrm{a}\left(\frac{\partial T_\mathrm{a}}{\partial t} + J(\psi_\mathrm{a}, T_\mathrm{a}) - \sigma\omega\frac{p}{R}\right) = -\lambda(T_\mathrm{a} - T_\mathrm{g}) + \epsilon_\mathrm{a}\sigma_\mathrm{B}T_\mathrm{g}^4 - 2\epsilon_\mathrm{a}\sigma_\mathrm{B}T_\mathrm{a}^4 + R_\mathrm{a} \tag{3}$$

$$\gamma_\mathrm{g}\frac{\partial T_\mathrm{g}}{\partial t} = -\lambda(T_\mathrm{g} - T_\mathrm{a}) - \sigma_\mathrm{B}T_\mathrm{g}^4 + \epsilon_\mathrm{a}\sigma_\mathrm{B}T_\mathrm{a}^4 + R_\mathrm{g} \tag{4}$$


where $T_g$ and $T_a$ are the ground and atmospheric temperature respectively. $\sigma$ is the static stability with $p$ as the pressure. $R$ is the gas constant for dry air. $\gamma_a$ is the heat capacity of the atmosphere for a 1000-hPa deep column where as $\gamma_g$ is the heat capacity of the active layer of the land for a mean thickness of 10 $m$ (Monin, 1986). $\lambda$ is the heat transfer coefficient between the land and atmosphere. $\sigma_B$ is the Stefan-Boltzmann constant and $\epsilon_a$ is the longwave emissivity of the atmosphere. $R_a$ is the

shortwave solar radiation directly absorbed by the atmosphere whereas $R_g$ is the shortwave solar radiation absorbed by the land.

As in Vannitsem et al. (2015) and De Cruz et al. (2016), quartic terms of the temperature equations are linearised. Upon nondimensionalization, the `qgs` framework represents the above equations as ordinary differential equations by projecting them on to a set of basis functions, a procedure which is also known as Galerkin expansion. We investigate the (2, 2) resolution

configuration of the modelfor the current study, which means that basis functions up to wavenumber 2 in each coordinate of





**Table 1.** Typical values of the model used for the study

| Parameter | value | Parameter | value |
|---|---|---|---|
| $a$ | 6371 km | $\sigma_B$ | $5.67 \times 10^{-8}$ Wm$^{-2}$K$^{-4}$ |
| $\pi L$ | 5000 km | $\gamma_a$ | $1.0 \times 10^7$ Jm$^{-2}$K$^{-1}$ |
| $H$ | 8.5 km | $\gamma_g$ | $1.6 \times 10^7$ Jm$^{-2}$K$^{-1}$ |
| $\phi_o$ | 50° N | $\lambda$ | 10 Wm$^{-2}$K$^{-1}$ |
| $n$ | 1.3 | $f_o$ | 0.0001032 s$^{-1}$ |
| $\epsilon$ | 0.76 | $R$ | 287.058 Jkg$^{-1}$K$^{-1}$ |
| $k_d$ | $1.2384 \times 10^{-5}$ s$^{-1}$ | $k_d'$ | $2.068 \times 10^{-6}$ s$^{-1}$ |
| $\sigma$ | $2.158 \times 10^{-6}$ m$^2$s$^{-2}$Pa$^{-2}$ | | |

Values of $\lambda, n, k_d$ are varied beyond the displayed value to study the model sensitivity.

the model are used. Consequently, the following list of 10 basis functions was used for the study:

$$
\begin{aligned}
F_1(x,y) &= \sqrt{2}\cos(y), \\
F_2(x,y) &= 2\cos(nx)\sin(y), \\
F_3(x,y) &= 2\sin(nx)\sin(y), \\
F_4(x,y) &= \sqrt{2}\cos(2y), \\
F_5(x,y) &= 2\cos(nx)\sin(2y), \\
F_6(x,y) &= 2\sin(nx)\sin(2y), \\
F_7(x,y) &= 2\cos(2nx)\sin(y), \\
F_8(x,y) &= 2\sin(2nx)\sin(y), \\
F_9(x,y) &= 2\cos(2nx)\sin(2y), \\
F_{10}(x,y) &= 2\sin(2nx)\sin(2y),
\end{aligned}
\tag{5}
$$

This configuration yields therefore a set of 30 variables, including 10 barotropic variables, 10 baroclinic variables, and 10 ground temperature variables. Note that as explained in the `qgs` documentation, the basis functions of the model can be easily
altered.

## 2.2 Model Parameters

Even though the model that we are using is a highly truncated spectral model, its comparability towards the original atmosphere and ground coupling is of great importance. Simulations produced by the model will be more meaningful if it characterizes *earthlike* properties. This can be obtained by tweaking and tuning the model parameters. In the present scenario, the parameters
used are derived from Reinhold and Pierrehumbert (1982), where they specifically estimated realistic parameter ranges that result in midlatitude terrestrial flow characteristics and regimes. The typical dimensional parameter values used for this study are displayed in Table 1.





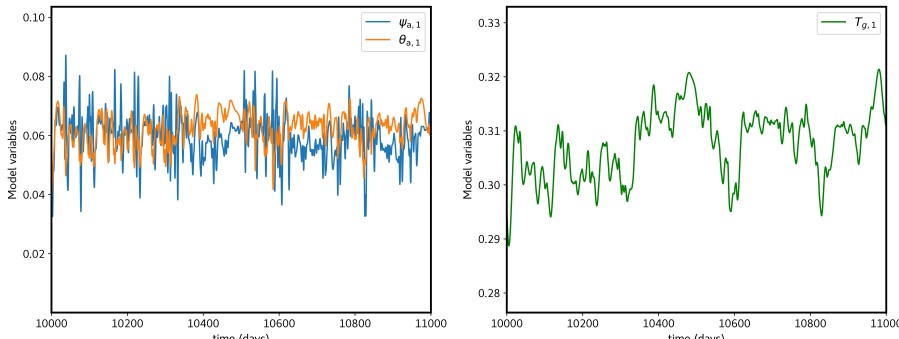

**Figure 1.** Temporal evolution of barotropic ($\psi_{a1}$) and baroclinic ($\theta_{a1}$) atmospheric streamfunction and the ground temperature ($T_{g1}$) for $C_g$ = 300 $Wm^{-2}$ and $k_d$ = 0.085 .

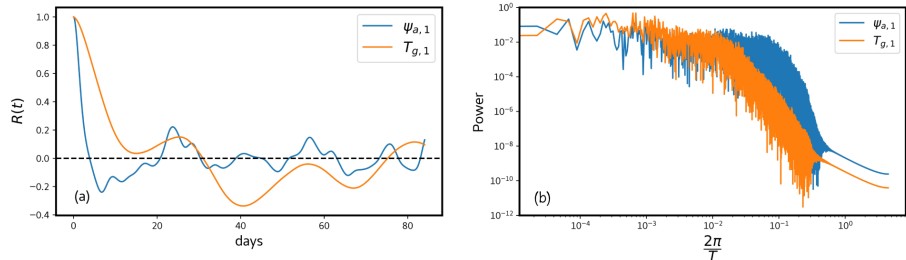

**Figure 2.** (a) Autocorrelation of barotropic ($\psi_{a1}$) atmospheric streamfunction and the ground temperature ($T_{g1}$) for $C_g$ = 300 $Wm^{-2}$ and $k_d$ = 0.085. (b) Powerspectrum of the same variables.

## 2.3 Model trajectories and mean fields

Figure 1 displays the time evolution of the first barotropic ($\psi_{a,1}$) and baroclinic ($\theta_{a,1}$) streamfunction modes of the atmosphere

and the ground temperature ($T_{g,1}$) for about 10 years starting after 10000 days of transient integration. Fluctuations are more erratic in the atmospheric part, which denotes its key role in the dynamics of the system. The variable representing the land component of the system is comparatively slower and less erratic. This difference suggests that the land component has a longer typical time scale than the atmosphere in this system.

Figure 2(a) emphasizes the observation above, where the autocorrelation of the first barotropic atmospheric mode and the

first ground temperature mode for $C_g$ = 300 $Wm^{-2}$ and $k_d$ = 0.085 are displayed. These, evaluated on a time series of 10000 days, helps us estimate the memory loss of the variables.

Indeed, the typical timescale of the processes at hand can be evaluated by the $e$-folding time which is the time beyond which the correlation has decreased by $1/e$. As expected the $e$-folding time of the atmospheric part is approximately 1.9 days which is comparitatively lower than that of the land part ($\approx 7.6$ days) indicating that the system is a multiscale model with a typical

timescale ratio of 10 for the specific parameter values considered.





Figure 2(b) is depicting the power spectra of modes $\psi_{a,1}$ and $T_{g,1}$ calculated by the Fourier transformation of the auto-correlation function using again timeseries of 10000 days. Atmospheric mode has a flat spectrum for lower frequencies and decays rapidly for higher frequencies. The spectrum for the ground mode is initially following the path of the atmosphere (lower frequencies up to 0.001) but starts to decay earlier which indicates more structured variabilities at lower frequencies than the atmosphere. The existence of a substantial continuous part in the spectrum is an indication of the complexity of the deterministic dynamics in time and suggests the presence of a chaotic dynamics (Arbabi and Mezić, 2017; Mezić, 2020).

## 3  Methodology

The objective of this research - besides introducing the basic equations, energy balance scheme and its sensitivity - is to explore the predictability of blocking and zonal weather regimes. By analyzing the peculiarities and the patterns of the land atmospheric coupled model, the study concludes that, when utilizing the parameters described in section 2.2, the system shows a qualitatively similar behaviour as the large scale actual atmosphere at midlatitudes (examples are illustrated in Appendix B). Moreover, varying the surface friction term, $k_d$, within the range of 0.06 to 0.12 yields numerous instances of realistic flow regimes, including blocking, zonal, and transition phases between these two states. To isolate between these flow regimes, a machine learning algorithm called Gaussian Mixture Clustering (GMC) is employed, which will be described in appendix A. After classifying the data, the average geopotential height at 500 hPa of each cluster is calculated to identify different flow regimes. The number of clusters is fixed to three using a trial and error method. Each flow regime is equally important due to its presence in the actual atmosphere. The cluster containing the lowest fraction of points in percentage being designated as the transition regime. The predictability horizon of each regime is evaluated by computing the inverse of the average largest local Lyapunov exponents of the clusters, which are calculated at each point of the attractor before clustering.

## 4  Regime stability and Lyapunov properties of the low order land atmospheric coupled model

Characterizing the instability properties of different flow regimes and their dependence with respect to important parameters are investigated as follows: The first part is exploring the sensitivity of the model about essential parameters which play a key role in structuring the output of the system. In the second part, the predictability properties of zonal, blocking and transition flow regimes using the Lyapunov exponents is investigated.

### 4.1  Stability properties of the model

Stability properties of the land-atmosphere coupled model at their equilibrium states are well depicted in Li et al. (2018). They also defined high-index equilibria and low-index equilibria based on the value of the streamfunction in the upper and lower atmospheric layer when the model solutions are equilibrium states. Even though stability of the model's equilibrium states is interesting and insightful, the actual atmosphere displays time-dependent solutions. Moreover, realistic atmospheric models





are chaotic and acutely sensitive to the initial conditions. Hence in this section, we investigated the stability properties of the land atmosphere model when its behavior is similar to *earthlike* situations with erratic dynamics.

Chaotic dynamical systems which exhibit sensitivity to the initial conditions can be qualitatively analysed by computing Lyapunov exponents and vectors.

## 4.2 Theory

Sensitivity to initial conditions is usually estimated using Lyapunov exponents. Let us consider an initial state, $\boldsymbol{x}(t_o) = \boldsymbol{x}_o$, a small perturbation $\delta\boldsymbol{x}_o$ is added to it which produces eventually a completely different trajectory. The dynamics of the error growth of the system can be linearized provided the perturbation is infinitesimally small as

$$\frac{d\delta\boldsymbol{x}}{dt} = \frac{\partial\boldsymbol{f}}{\partial\boldsymbol{x}}\bigg|_{\boldsymbol{x}(t)}\delta\boldsymbol{x} \tag{6}$$

and its solution is

$$\delta\boldsymbol{x}(t) = \mathbf{M}(t, \boldsymbol{x}(t_o))\delta\boldsymbol{x}(t_o) \tag{7}$$

where $\mathbf{M}$ is known as the resolvent or propagator matrix. The Euclidean norm of the error can be computed as

$$E_t = |\delta\boldsymbol{x}(t)|^2 = \delta\boldsymbol{x}(t)^T\delta\boldsymbol{x}(t)$$

$$E_t = \delta\boldsymbol{x}(t_o)^T\mathbf{M}(t, \boldsymbol{x}(t_o))^T\mathbf{M}(t, \boldsymbol{x}(t_o))\delta\boldsymbol{x}(t_o) \tag{8}$$

hence indicating that the error growth is provided by the eigenvalues of $\mathbf{M}^T\mathbf{M}$, where $\mathbf{M}^T$ denotes the transpose of the
resolvent matrix $\mathbf{M}$. By the multiplicative ergodic theorem of Oseledec (Eckmann and Ruelle, 1985; Kuptsov and Parlitz, 2012), a double limit is considered with perturbation amplitude going to 0 and time going to infinity. The logarithm of the eigenvalues of matrix $(\mathbf{M}^T\mathbf{M})^{2(t-t_o)}$ within these limits, which are known as Lyapunov exponents, quantifies the divergence of the initially close trajectories. The complete set of Lyapunov exponents that are usually represented in decreasing order constitutes the Lyapunov spectrum. Different types of Lyapunov vectors existed based on the method of calculation like forward
Lyapunov vectors, backward Lyapunov vectors and covariant Lyapunov vectors whose properties are described in details in (Kuptsov and Parlitz, 2012; Legras and Vautard, 1996).

In the current study, we are using backward Lyapunov vectors (BLVs) which are obtained by considering the eigenvalues of the matrix $(\mathbf{M}^T\mathbf{M})^{2(t-t_o)}$ by taking initial state $t_o \to -\infty$. Several numerical techniques exist for the calculation of the Lyapunov exponents. The most common method is using Gram-Schmidt orthonormalization (Shimada and Nagashima, 1979;
Parker and Chua, 2012): A set of orthonormal random vectors are propagated in the tangent space of the trajectory, according to equation (6), frequently re-orthonormalizing this basis to avoid the collapse of all the vectors towards the most unstable direction which is associated with the largest Lyapunov exponent. After a transient, these vectors provide the BLVs and concurrently



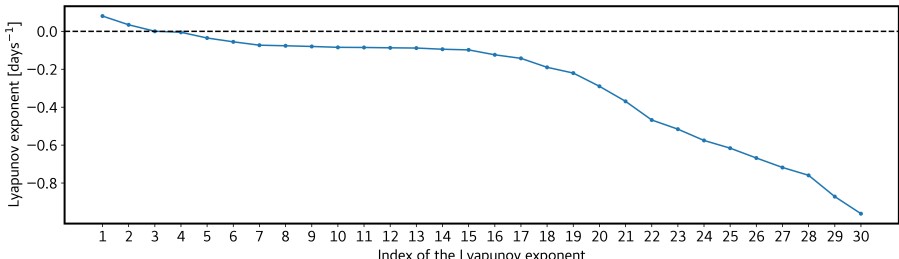

**Figure 3.** Lyapunov spectra of the land atmospheric coupled model for $C_g = 300 \ Wm^{-2}$ and $k_d = 0.085$. The values of the Lyapunov exponents are given in $\mathrm{days}^{-1}$ .

the sought Lyapunoc exponents. The complete set of these vectors give the full picture of the instability of the trajectory in phase space.

### 4.3 Lyapunov spectra and averaged variance of the model

Figure 3 displays the Lyapunov spectrum of the land-atmosphere coupled model when $C_g = 300 \ Wm^{-2}$ and $k_d = 0.085$. All other parameters are provided in Table 1. The system has 3 positive, 1 zero and 26 negative Lyapunov exponents. Similarly to the ocean-atmospheric coupled model (Vannitsem et al., 2015; Vannitsem and Lucarini, 2016; Vannitsem, 2017), the spectrum contains a set of Lyapunov exponents forming a plateau close to 0, but the amplitude of the Lyapunov exponents around this plateau is however quite substantial as compared to the coupled ocean-atmosphere model. This plateau is expected to be associated with the presence of the land whose typical time scales of variability are slower than the atmosphere.

The coupling between land and atmosphere plays a key role in the behaviour of the model. Hence quantifying the extent of this coupling and its instabilities is essential for interpreting the properties of the model. Therefore, the averaged variance of the Lyapunov vectors is displayed in figure 4 to elucidate this information along each variable.

The first ten variables represent the barotropic streamfunction of the atmosphere, the next ten represent the baroclinic streamfunction of the atmosphere, and the last ten represent the ground temperature. The variance of the BLVs is primarily residing in the atmospheric part, indicating that this part primarily contributes to the system's dynamics. It should also be noted that the atmospheric part includes the most unstable BLVs (corresponding to the first two Lyapunov exponents) as well as the most stable BLVs (21 to 30 corresponds to large negative LEs).

Conversely, variance is predominantly projected on the temperature variables of the ground part for the BLVs 3 to 20. These BLVs are associated with the plateau formed by the near-zero LEs visible in Fig. 3. The observation of comparable variance in both the atmospheric and ground parts describes (horizontally) the coupling in the model, which is thus represented for BLVs 3-7. For BLVs 21 to 30, variance projection is almost non-existent in the ground part, indicating that the ground part makes almost no contribution to the stabilization of the system.

Figure 5 displays Lyapunov spectra calculated for various energy input levels ($C_g$) ranging from 300 to 400 $Wm^{-2}$.



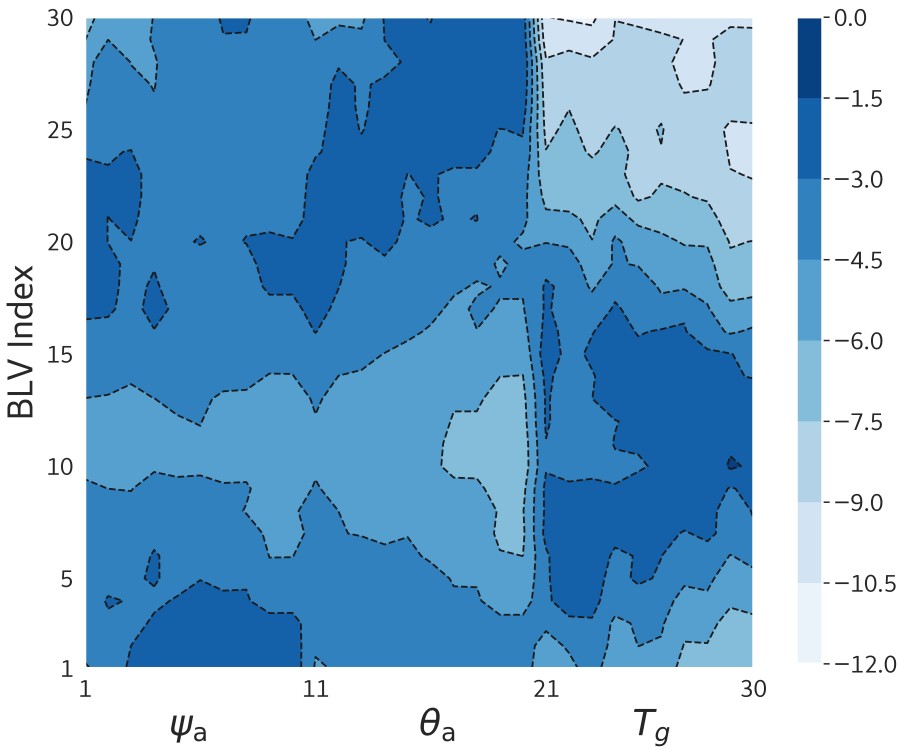

**Figure 4.** Values of the time-averaged and normalized variance of the BLVs as a function of the variables of the model ($log_{10}$ scale). The 20 first modes correspond to the variables of the atmosphere and the next 10 ones correspond to the temperature of the ground. Parameters' value: $C_g = 300$ Wm$^{-2}$ and $k_d = 0.085$. The Euclidean norm is used for all BLVs, and their squared norm is normalized to 1

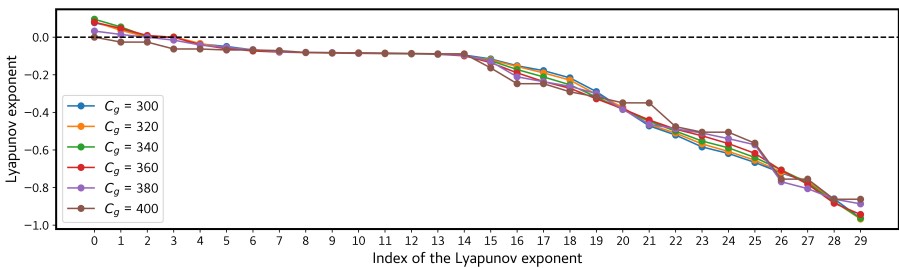

**Figure 5.** Lyapunov spectra of the land atmospheric coupled model for different values of $C_g$. Each color represents corresponding $C_g$ values used for calculating the spectrum. The values of the Lyapunov exponents are given in days$^{-1}$

When $C_g$ is lower, the model exhibits chaotic behavior, indicated by two positive exponents, one zero exponent, and 26 negative exponents. Surprisingly, the amount of incoming shortwave radiation doesn't significantly affect the spectrum between $C_g$ values of 300 and 360 Wm$^{-2}$.

However, as $C_g$ increases, the system shifts to periodic behavior, characterized by one zero exponent and 29 negative

255 Lyapunov exponents, before switching back to chaos.




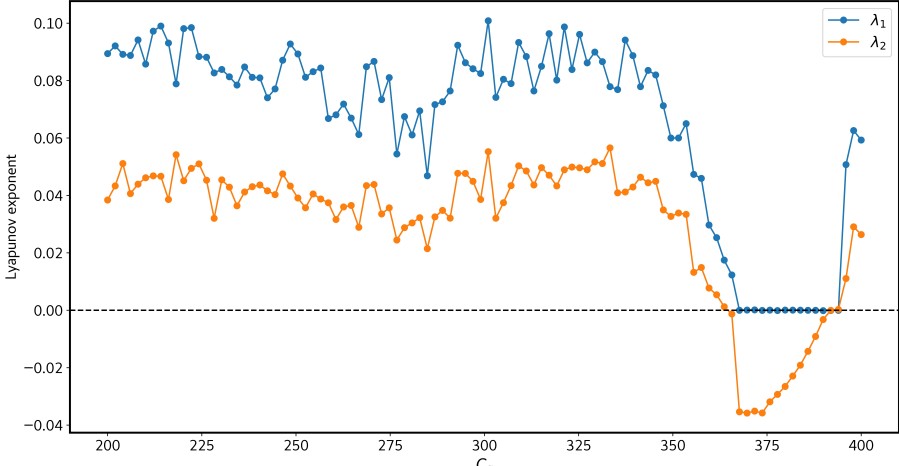

**Figure 6.** First and second Lyapunov exponents of the land atmospheric coupled model for different values of $C_g$. blue line represents first and orange line represents second Lyapunov exponents respectively. The values of the Lyapunov exponents are given in days$^{-1}$

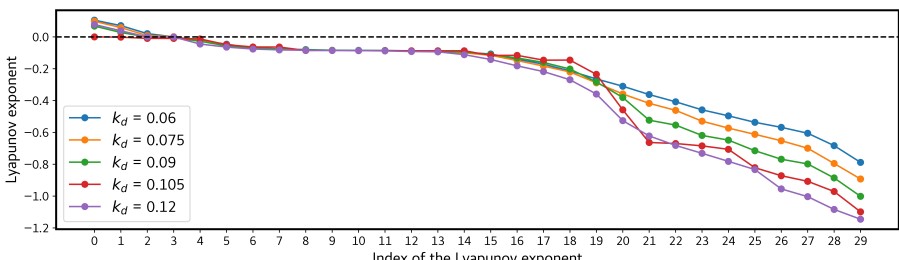

**Figure 7.** Lyapunov spectra of the land atmosphere coupled model for different values of $k_d$. Each color represents corresponding $k_d$ values used for calculating the spectrum. The values of the Lyapunov exponents are given in days$^{-1}$.

This is illustrated in Figure 6 where the first and second Lyapunov exponents are positive for the system for the lower values of $C_g$ and then system enters in to a periodic window for the $C_g$ values and revert back again to chaotic behaviour later.

The surface friction $k_d$ is also affecting stability of the system to a great extent. In order to investigate the sensitivity of the model towards $k_d$, Lyapunov spectra were drawn with the parameter values exhibited in table 1 with different $k_d$ values as displayed in Fig. 7.

The non-dimensional $k_d$ values depicted in the figure were obtained from Reinhold and Pierrehumbert (1982), where they asserted that flow regimes generated using these values exhibit realistic midlatitude terrestrial properties. Among the system configurations with $k_d$ values of 0.06, 0.075, 0.09, and 0.012, there are 3 positive, 1 zero, and 26 negative Lyapunov exponents. Up to the point of plateau formation, all these spectra display similar behavior. Note that the spectrum corresponding to $k_d$ = 0.12 exhibits notably strong negative Lyapunov exponents, while the spectrum for $k_d$ = 0.06 demonstrates comparatively weaker negative values, as expected from the increased associated dissipation.





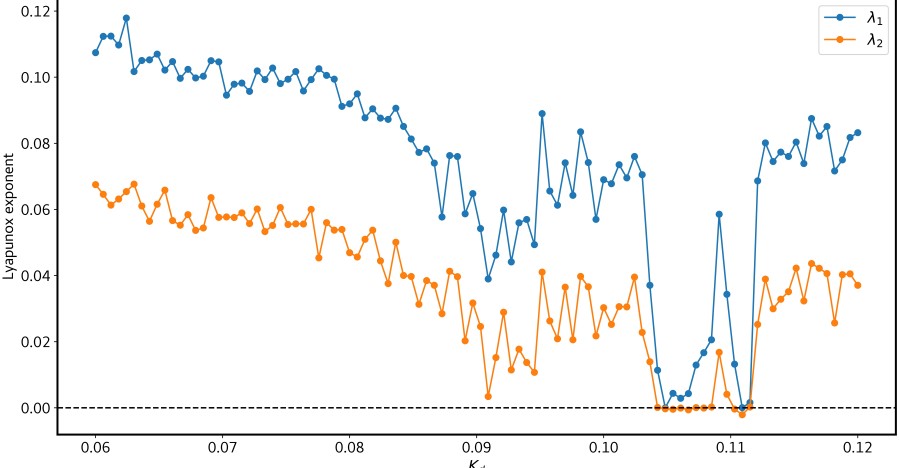

**Figure 8.** First and second Lyapunov exponents of the land-atmosphere coupled model for different non-dimensional values of $k_d$. Blue line represents the first and orange line represents the second Lyapunov exponents, respectively.$C_g = 300$ Wm$^{-2}$. The values of the Lyapunov exponents are given in days$^{-1}$

.

From Figure 4, we identified that BLVs 20 - 30 are actually depicting the variance of temperature of the atmosphere. Given the significant variation in BLVs 20 - 30 within the current context, it can be inferred that the atmospheric temperature gradient, and hence baroclinic instability is becoming weaker and the system is stabilizing due to the changes in the surface friction.

As thoroughly explained in section 4.2, the largest Lyapunov exponent serves as an indicator of the system's highest degree of instability, while the second positive Lyapunov exponent represents the second most unstable characteristic, and so forth. In Fig. 8, the results demonstrate that at lower values of $k_d$, the system exhibits a highly chaotic nature. However, as we increase $k_d$ towards higher values, the system stabilizes, revealing a periodic window. This can also be seen with the Lyapunov spectrum for $k_d = 0.105$ in Fig. 7.

Subsequently, with further increment in $k_d$, the system returns to a more chaotic behavior, illustrating the complex role of dissipative features.

The primary interaction between land and atmosphere in the model is facilitated through heat exchange denoted by $\lambda$. Therefore, it is crucial to understand and analyze how alterations in the heat exchange mechanism influence the behavior of the model. This is illustrated in figure 9.

When $\lambda$ is set to zero, the system displays a periodic behavior characterized by one zero and 29 negative Lyapunov exponents. However, for all other non-zero $\lambda$ values, the system exhibits chaos with 3 positive exponents, 1 zero, and 26 negative Lyapunov exponents. Smaller values of $\lambda$ (e.g., $\lambda = 10, 25,$ and $50$) yield a smooth spectrum with a plateau which is similar to the typical Lyapunov spectrum encountered before. Note that with higher $\lambda$ values, an anomalous bend is observed in the spectrum, specifically from the 20th exponent onward, indicating unrealistic stability. It is also interesting to note that the spectrum associated with the intersection of the land part (between 20th and 21st exponent) is becoming steeper with the increase of $\lambda$.





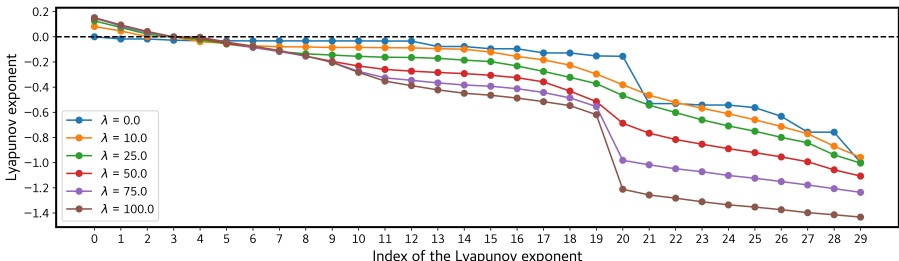

**Figure 9.** Lyapunov spectra of the land atmospheric coupled model for different values of $\lambda$. Each color represents corresponding $\lambda$ values used for calculating the spectrum. The values of the Lyapunov exponents are given in days$^{-1}$.

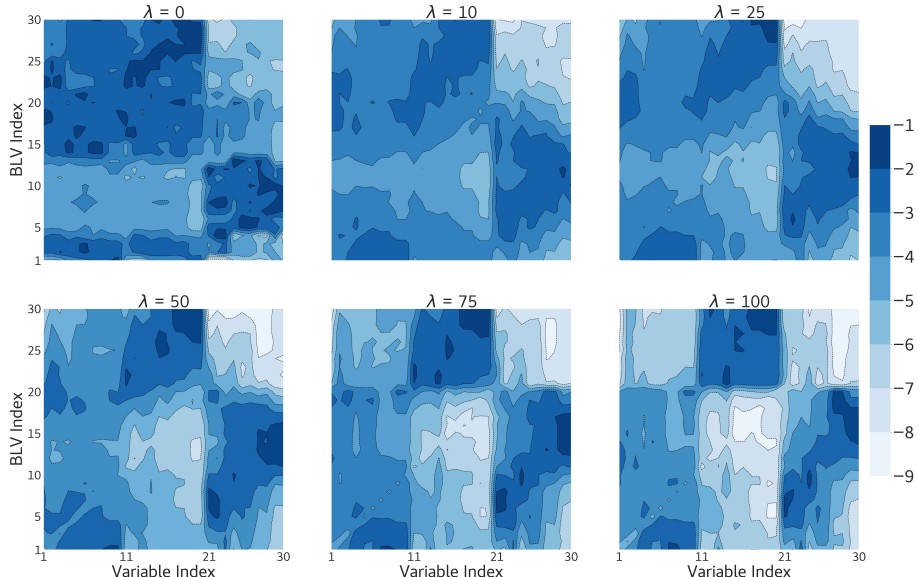

**Figure 10.** Values of the time-averaged and normalized variance of the BLVs as a function of the variables of the model ($log_{10}$ scale) for different $\lambda$ values in Wm$^{-2}$K$^{-1}$. The 20 first modes corresponds to the variables of the atmosphere and the next 10 ones corresponds to the temperature of the ground. Parameters' value: $C_g = 300$ Wm$^{-2}$ and $k_d = 0.085$. The Euclidean norm is used for all BLVs, and their squared norm is normalized to 1.

The increased heat exchange leads to a reduced temperature difference between the atmosphere and the ground, thereby giving rise to this particular situation. This can be further explained by the averaged variance of Lyapunov exponents for different values of heat exchange $\lambda$ in Figure 10.

A clear separation between land and atmosphere exists at $\lambda = 0$, when there is no exchange of heat between them, resulted in to respective spectrum. For $\lambda = 10$, 25, and 50, the distribution remains similar, with variances concentrated predominantly in the atmospheric component for the first three BLVs and the last ten BLVs. BLVs 3-20 represent a plateau-like pattern, indicating a coupling between the atmosphere and the ground. Despite variance distribution, it's notable that there is a uniform spread rather than strong concentration or absence of variance. For $\lambda = 75$ and 100, the variance distribution becomes compartmental-





ized. Variance is now concentrated within the first 20 BLVs for the barotropic streamfunction and ground temperature, while
the last 10 BLVs primarily represent atmospheric temperature or baroclinic streamfunction. In contrast to earlier cases, there
is a clear absence of variance within the middle portion, specifically confined to BLVs 20-30. This compartmentalized energy
distribution leads to a characteristic jump in the corresponding spectrum.

## 5 Predictability properties of zonal, blocking and transition flow regimes

Upon concluding our investigation of the land-atmosphere coupled model's utility in studying low-frequency variability in
the atmosphere, we have identified zonal, blocking, and transition regimes concerning different $k_d$ values, as explained in
section 3. For the range of $k_d$ values between 0.06 and 0.12, the lower values (0.06 to 0.075) encompass all three regimes:
zonal, blocking, and transition. As we progress from 0.08 onward, we observe two blocking regimes and a transitional regime
between them until $k_d = 0.10$. Subsequently, the system exhibits periodic behavior. At $k_d = 0.115$, two blocking regimes are
observed, and further, at $k_d = 0.12$, three flow regimes are identified. These findings demonstrate the model's capability to
capture various flow regimes and their transitions based on the selected range of $k_d$ values.

The predictability horizon of the zonal regimes is found to be longer compared to the blocking regimes in the range of $k_d$
values where all three regimes coexist ($k_d = 0.06$ to 0.075). This implies that the blocking regimes exhibit higher instability
in agreement with previous findings(Schubert and Lucarini, 2016; Faranda et al., 2016, 2017; Lucarini et al., 2016). The
transition regimes, on the other hand, show notably lower predictability in comparison to the other regimes. Within the interval
of $k_d$ values encompassing only two blocking regimes and a transition regime ($k_d = 0.08$ to 0.10), both blocking regimes
display significantly different predictability horizons. At the outset ($k_d = 0.08$), they demonstrate a predictability difference
of approximately 1 day. Subsequently, this difference increases and reaches about 10 days at $k_d = 0.095$. However, it then
decreases again to a difference of 1 day when $k_d = 0.10$. Moreover, it is observed that the stability of the transition regime
surpasses that of one of the blocking regimes. This indicates the occurrence of a qualitative change in the predictability of the
blocking regimes within this particular range of $k_d$ values.

In these instances, it is noteworthy that situations characterized by lower predictability or significant instability tend to occur
when atmospheric blocking takes place on the western side of topographical features. Conversely, when blocking occurs on the
eastern side of such topography, it exhibits greater stability and a longer predictability horizon. This observation draws parallels
with real-world scenarios, such as the persistence of North Pacific blocking patterns (Breeden et al., 2020; Kim and Kim, 2019).
The morphology of the identified blocking events bears resemblance to North Pacific blocks, where a high-pressure system is
situated either to the west or east of the underlying topography. These locations correspond to the windward and leeward sides
of the mountain ranges in the model.

The system is entering a periodic window after $k_d = 0.10$. Then it again become chaotic with 2 blocking regimes and a
transition regime and later on with 3 distinct regimes. Figure 11 depicts all the findings obtained from this study.



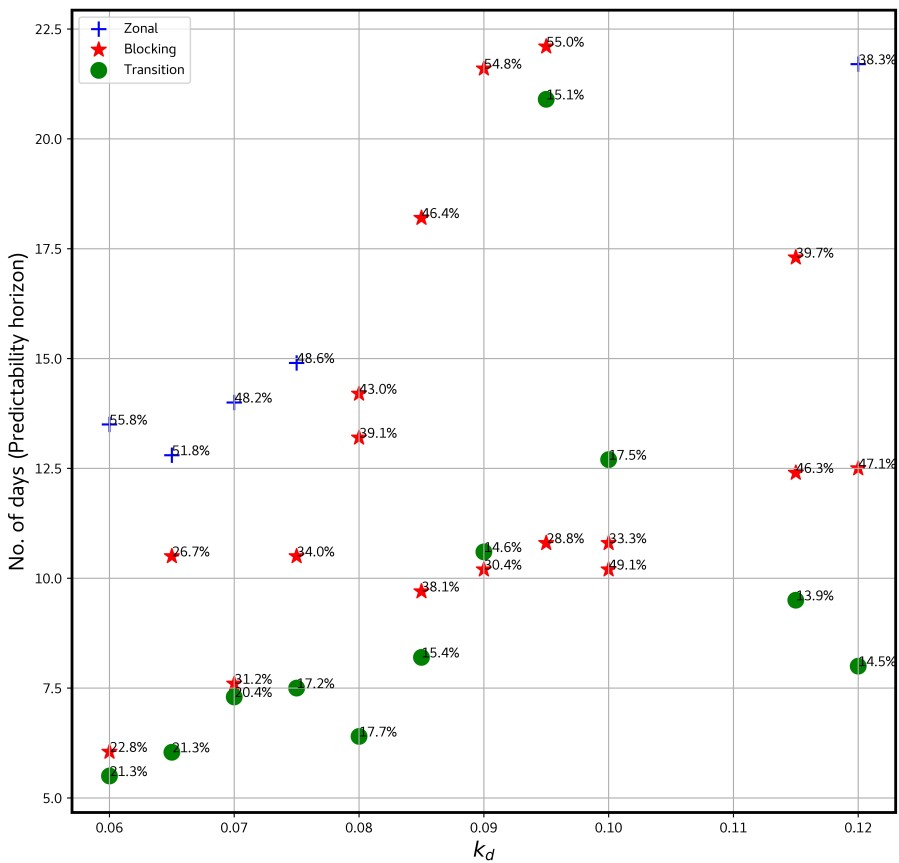

**Figure 11.** Predictability horizon of zonal, blocking and transition flow regimes for different non-dimensional $k_d$. The numbers represents the percentage of the total number of points included in the respective regime. Predictability horizon is depicted in days

## 6 Impact of model resolution

As per Cehelsky and Tung (1987), resolution of the model can affect the output in various ways. For instance, the nonlinear interaction between the modes (as number of modes increases, resolution of the system increases) is quite altered if different number of modes are considered, resulting in entirely different dynamics for the same set of parameters. Hence, analysis of high resolution runs and its Lyapunov properties is inevitable. In this section, the system is ran with a higher resolution configuration (5,5) with 55 modes being used for both the atmospheric and land part, giving a system with a total of 165 variables. Parameter values are the same as that of the earlier analysis listed in Table 1. Figure 12 is depicting the Lyapunov spectrum and Figure 13 is representing the time-averaged variance of the BLVs of the high resolution run.

The Lyapunov spectrum exhibits structural similarity when compared to the low-resolution spectra, except that more positive exponents are present. Specifically, it comprises 12 positive, one zero, and 152 negative exponents. Notably, there is a distinctive plateau observable within the range of 20 to 75, which is a characteristic feature attributed to the interaction between the land



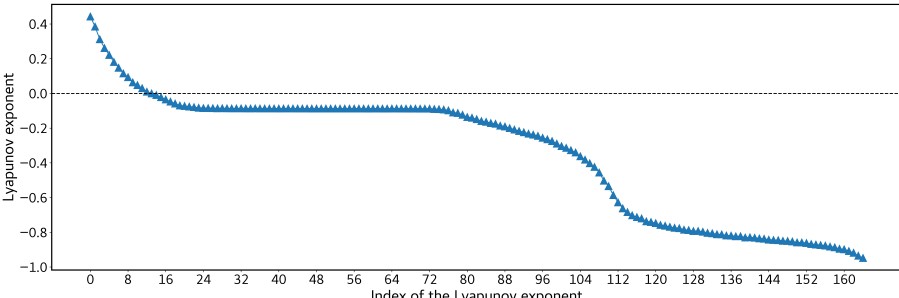

**Figure 12.** Lyapunov spectrum of the land-atmosphere coupled model for $C_g = 300$ Wm$^{-2}$ and $k_d = 0.085$ in (5,5) model configuration. The values of the Lyapunov exponents are given in days$^{-1}$.

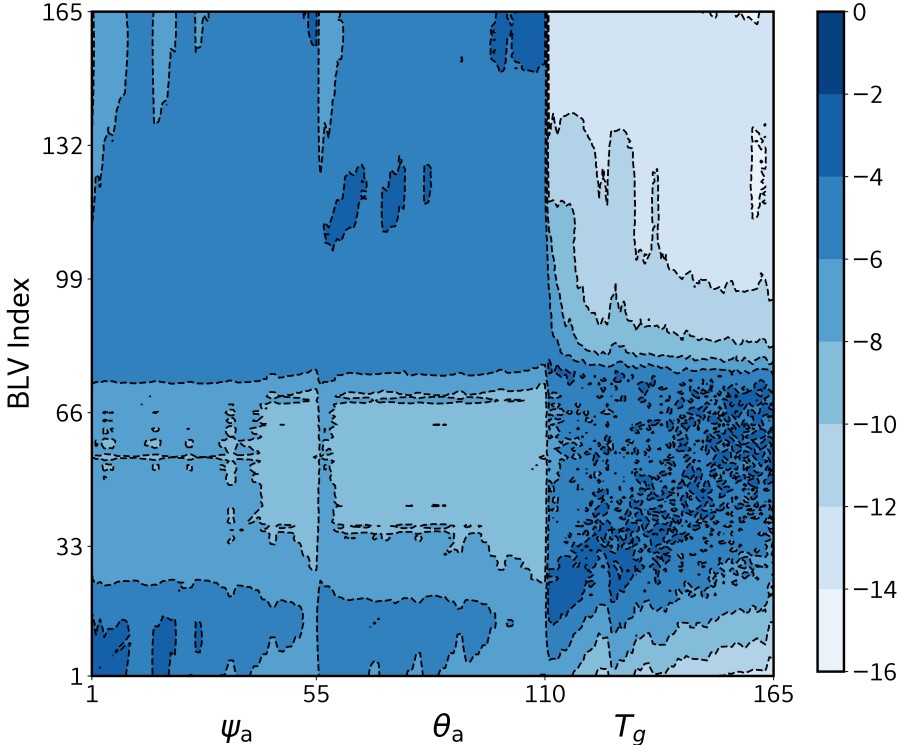

**Figure 13.** Values of the time-averaged and normalized variance of the BLVs as a function of the variables of the model ($log_{10}$ scale) in (5,5) model configuration. The 110 first modes corresponds to the variables of the atmosphere and the next 55 ones corresponds to the temperature of the ground. Parameters' value: $C_g = 300$ Wm$^{-2}$ and $k_d = 0.085$. The Euclidean norm is used for all BLVs, and their squared norm is normalized to 1.

and the atmosphere in the model. This plateau phenomenon arises due to the disparity in timescales resulting from the intricate interplay between land and atmosphere dynamics.





In Fig. 13, higher variances in the atmosphere are observed on first 40 BLVs and also at the BLVs greater than 70 indicating that the most chaotic and stable dynamics are resulting from the atmospheric component. Furthermore, it is noteworthy that a

greater concentration of variance is observed to be shifted towards the variables that specifically represent the ground temperature, particularly within the range of BLVs 20 to 75. This shift is a key factor contributing to the formation of a plateau within the spectrum in question. It becomes evident that there is a distinct compartmentalization between each set of variables, including the barotropic, baroclinic, and ground temperature variables, which is more pronounced when compared to the variance illustration at lower resolutions.

The outcome of the clustering analysis has delineated two clearly defined patterns: one marked by zonal flow and the other by instances of blocking. The intermediate pattern that once existed between these two regimes is now absent. Notably, both the zonal and blocking events exhibit an identical predictability horizon, specifically spanning a period of two days. This feature is contrasting with what is found at the lower resolution and also in the current literature on this topic. This aspect is worth investigating further in the future by exploring other sets of parameters and other resolutions.

## 7   Discussions

This study focused on characterizing the variability and instability properties of different flow regimes and their dependence on important parameters in an idealized coupled model, namely the quasi-geostrophic land atmosphere coupled model. The investigation also aimed at exploring the predictability of zonal, blocking, and transition flow regimes using Lyapunov exponents.

The analysis revealed that the model is less sensitive to variations in meridional differences in solar heating absorbed by the land, represented by $C_g$. Based on observations, a fixed value of 300 $\text{Wm}^{-2}$ for $C_g$ was chosen for further analysis. The study found that the model's stability is significantly affected by surface friction $k_d$. Different values of $k_d$ were explored, and it was observed that at lower $k_d$ values, the system exhibits chaotic behavior, while at higher $k_d$ values, periodic windows alternate with chaotic behaviors. Within this range, the systems solutions meander between various flow regimes, including

zonal, blocking, and transition.

The heat exchange mechanism, represented by $\lambda$, was also analyzed, and it was found that when $\lambda$ is set to zero, the system displays periodic behavior, while for non-zero $\lambda$ values, the system exhibits considerable chaos. Overall, the model demonstrated the capability to capture various flow regimes and their transitions based on the selected range of $k_d$ values, providing insights into the potential behavior of the atmosphere.

The predictability properties of three distinct flow regimes were investigated: zonal, blocking, and transition, which were the fundamental components of low-frequency variability (LFV) in the model. The predictability horizon of the zonal regimes was found to be longer compared to the blocking regimes which is consistent with earlier results (Schubert and Lucarini, 2016; Faranda et al., 2016, 2017; Lucarini et al., 2016), when all three regimes coexist. The transition regimes showed notably lower predictability compared to the other regimes. Within a specific range of $k_d$ values, the blocking regimes displayed different

predictability horizons, indicating the occurrence of a qualitative change in the predictability of the blocking regimes.



Weather patterns that involve atmospheric blocking to the west of a given topographical feature tend to have reduced predictability and show instability when contrasted with blocking occurrences situated to the east of such topographical elements. This finding aligns with actual meteorological occurrences, such as the persistence of North Pacific blocking patterns (Breeden et al., 2020; Kim and Kim, 2019). The shape and characteristics of the identified blocking events closely resemble North Pacific
blocks, where a high-pressure system exists either on the western or eastern side of the underlying topography. In the physical world, these positions correspond to the windward and leeward sides of mountain ranges composed of rocky terrain. Despite `qgs` models being regarded as less comprehensive, their utilization in this study allows for a more relevant impact, akin to real-world situations as described above.

Upon increasing the model resolution, Lyapunov properties exhibited a remarkable resemblance to those observed at lower
resolutions. However, the distribution of points on the attractor gave rise to two distinct clusters, delineating the blocking and zonal regimes, thereby extinguishing the potential for the transition regime. Notably, both the blocking and zonal regimes displayed a predictability horizon limited to 2 days.

In the backdrop of rising global temperatures and the escalation of climate extremes, comprehending the intricate dynamics governing atmospheric blocking occurrences and their predictability becomes paramount, given that blocking events are
consistently linked to extreme weather phenomena. The impact of climate change can also be explored in the current study by modifying the emissivity of the atmosphere. This will be explored in the future.

The knowledge acquired through this study holds potential significance for climate and weather prediction models, contributing to the advancement of our understanding of the crucial atmospheric dynamics shaping the Earth's climate system. In future investigations, we will assess the influence of the same parameters within more complex models, enabling us to conduct
comparisons that will aid in identifying alterations in land-atmosphere interactions as atmospheric complexity intensifies. This undertaking will further our comprehension of how the interaction between land and the atmosphere evolves with increasing intricacies in atmospheric systems and how it affect the predictability of the weather regimes.

*Code availability.* The code used to obtain the results is a new version (v0.2.7) of `qgs` (Demaeyer et al., 2020) that was recently released on GitHub.

**Appendix A: Gaussian Mixture Clustering (GMC)**

Gaussian Mixture Clustering (GMC) is a prevalent unsupervised machine learning technique utilized for partitioning data points into clusters by modeling their underlying distribution. The method assumes that the data arises from a combination of multiple Gaussian distributions. Each cluster is characterized by a Gaussian component, and the primary objective is to accurately estimate the parameters of these components to optimally describe the data. It comprises several steps.





- **Initialization:** GMC starts by randomly selecting $K$ initial centers (means) $\mu_i$ for each cluster. Additionally, it initializes the covariance matrices $\Sigma_i$ and the mixing coefficients $\pi_i$, which represent the probabilities of data points belonging to each cluster, where $i = 1, 2, ..., K$. The mixing coefficients must sum up to 1, and each must be between 0 and 1.

- **Expectation - Maximization (EM) Algorithm:** GMC employs the EM algorithm (Dempster et al., 1977) to iteratively estimate the parameters of the Gaussian components. The algorithm comprises two steps, the Expectation step (E-step) and the Maximization step (M-step).

  - **Expectation Step (E-Step):** During this step, the algorithm computes the responsibility ($\gamma_{i,j}$) of each data point $x_j$ for each cluster $i$. The responsibility represents the probability that data point $x_j$ belongs to cluster $i$, given the current parameters. This is calculated using Bayes' theorem:

$$\gamma_{i,j} = \frac{\pi_i \cdot \mathcal{N}(x_j; \mu_i, \Sigma_i)}{\sum_{k=1}^{K} \pi_k \cdot \mathcal{N}(x_j; \mu_i, \Sigma_i)} \tag{A1}$$

  where $\gamma_{i,j}$ is the responsibility of cluster $i$ for data point $x_j$, $\pi_i$ is the mixing coefficient for cluster $i$. $\mu_i$ and $\Sigma_i$ are the mean and covariance matrix for cluster $i$, respectively. $\mathcal{N}(x_j; \mu_i, \Sigma_i)$ is the Gaussian probability density function for data point $x_j$ with mean $\mu_i$ and covariance $\Sigma_i$.

  - **Maximization Step (M-Step):** In this step, the algorithm updates the parameters of the Gaussian distributions (mean, covariance, and mixing coefficients) based on the responsibilities calculated in the E-step:

$$\text{New mean}, \mu_i = \frac{\sum_{j=1}^{N} \gamma_{i,j} \cdot x_j}{\sum_{j=1}^{N} \gamma_{i,j}} \tag{A2}$$

$$\text{New covariance matrix}, \Sigma_i = \frac{\sum_{j=1}^{N} \gamma_{i,j} \cdot (x_j - \mu_i) \cdot (x_j - \mu_i)^T}{\sum_{j=1}^{N} \gamma_{i,j}} \tag{A3}$$

$$\text{New mixing coefficient}, \pi_i = \frac{1}{N} \sum_{j=1}^{N} \gamma_{i,j} \tag{A4}$$

  where $N$ is the number of data points. $\mu_i$, $\Sigma_i$, and $\pi_i$ are the updated mean, covariance matrix, and mixing coefficient for cluster $i$, respectively. $\gamma_{i,j}$ is the responsibility of cluster i for data point $x_j$ (computed in the E-step). $x_j$ is the $j$-th data point.



- **Convergence:** The E-step and M-step are repeated iteratively until the algorithm converges. Convergence happens when the change in the likelihood of the data between iterations becomes very small or when a predefined number of iterations is reached.

- **Cluster Assignment:** Once the algorithm converges, each data point is assigned to the cluster with the highest probability (highest responsibility).

- **Number of Clusters ($K$):** The number of clusters, $K$, is typically determined either by the user based on prior knowledge or by using techniques like the Bayesian Information Criterion (BIC) or cross-validation to find the optimal number of clusters.

In our study, we used cross-validation method and decided $K$ as 3 for obtaining realistic results. Further explanation regarding the clustering algorithm can be obtained from recent literature on that subject (Hastie et al., 2009; Bishop and Nasrabadi, 2006; Dempster et al., 1977).

## Appendix B: Examples of blocking and zonal patterns evolved from the study

As previously indicated, employing the parameter configuration outlined in Table 1, we have successfully generated *earthlike* flow patterns. These flow regimes are visually represented in figures B2 and B1, corresponding to different values of the parameter $k_d$. Specifically, for lower values of $k_d$, we observe the coexistence of zonal, blocking, and transitional flow regimes, as depicted in figure B1. Conversely, when $k_d$ assumes higher values, we observe the emergence of two blocking flow regimes along with a transition regime, as illustrated in figure B2.





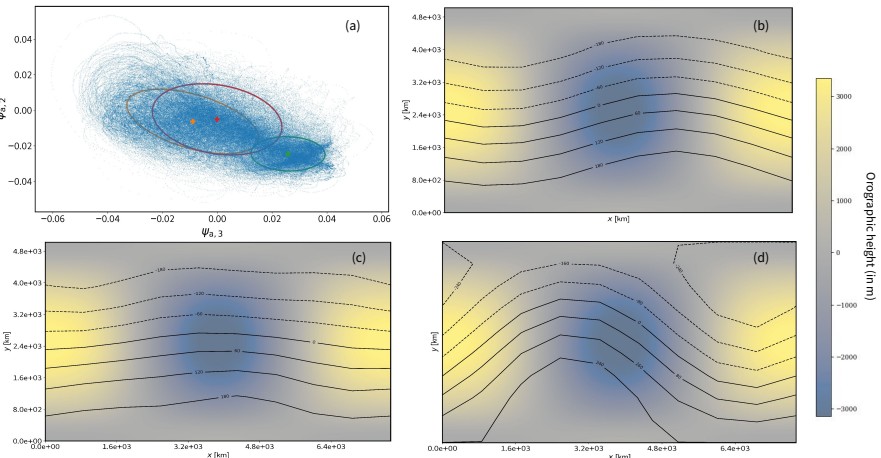

**Figure B1.** Phase space dynamics of the model projected on the $(\psi_{a,3}, \psi_{a,2})$-plane is shown in panel (a) for $k_d = 0.065$. Gaussian mixture clusters covariance are represented with orange, green and red ellipsis. Orange cluster is associated with a zonal regime which can be identified by the geopotential height at 500 hPa in panel (b). Red and green clusters are respectively transition and blocking regimes attributed in panel (c) and (d), also at 500hPa geopotential height. The orographic profile of the domain is depicted in panels (b), (c) and (d).

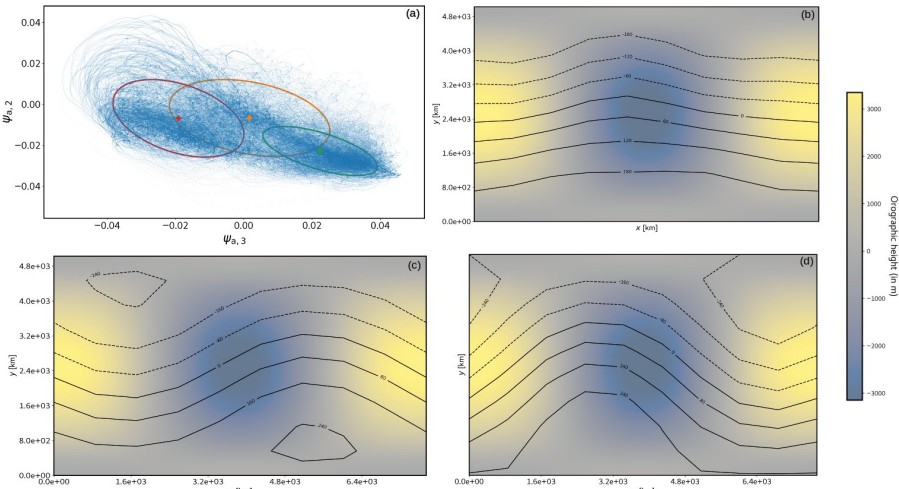

**Figure B2.** Same as Fig. B1 but for $k_d = 0.08$. Here the red (panel (c)) and green clusters (panel (d)) are both blocking regimes and the orange one (panel (b)) is a transition regime.



*Author contributions.* AKX contributed to conceptualisation, method development, method implementation and data analysis and writing and visualisation. JD contributed to conceptualisation, model and method development, and text improvements. JD is also the developer of `qgs` land-atmosphere coupled model used for the study. SV contributed to supervision, conceptualisation and writing.

*Competing interests.* The contact author has declared that none of the authors has any competing interests.

445 *Acknowledgements.* We would like to thank Thomas Gilbert, Francesco Ragone and Oisín Hamilton for the helpful discussions.This project received funding from the European Union's Horizon 2020 research and innovation programme under Marie Skłodowska-Curie grant No. 956396.



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
