# Peer review of "Variability and Predictability of a reduced-order land atmosphere"

_EGUsphere, 2023_

## Referee Comment (RC2)

*Reviewer's Comment* on
**Variability and Predictability of a reduced-order land atmosphere coupled model**
by Anupama K Xavier, Jonathan Demaeyer, and Stéphane Vannitsem

*General*. This paper presents a coupled land-atmosphere model that is applied to midlatitude atmospheric variability and predictability. The model itself is formulated in a flexible Python framework called qgs (Demaeyer et al., *J. Open Source Software*, 2020) that adds surface features, oceanic or terrestrial, to a two-layer quasi-geostrophic model in a periodic beta-channel. It seems to be the first application of qgs to coupling with a land surface. The Vannitsem group in Brussels has earned already quite a reputation with its modular arbitrary-order ocean-atmosphere model (MAOOAM: De Cruz et al., *Geoscientific Model Development*, 2016) that has been used worldwide for nonlinear studies of the climate in several configurations and on different time scales.

   The present paper examines the effects of the surface features on the number and stability of distinct atmospheric midlatitude regimes, including zonal, blocking and transient ones. The key tools are the backward Lyapunov spectrum and the associated vectors, abbreviated as BLVs, which are studied as a function of the coupling coefficients between the atmosphere and the ground, namely those that scale the momentum and heat exchanges. The model has a fairly low order, with a total of 30 spectral variables, ten each for the barotropic and baroclinic components of the atmosphere, and ten more for the ground temperature field. Despite this low order, the spatial features of the blocking and zonal regimes in model simulations are rather realistic; see Appendix B.

   The most interesting finding of the paper, to my mind, is the greater predictability of zonal flows, when this regime coexists with the blocked one for the same parameter values. This finding seems to contradict both the prevalence of zonal flows over blocked ones in observations and the widely held belief in the community that it is blocked flows that are stabler. More on this in the first major comment below.

   Overall, the paper should be accepted and published pretty much in the present form. Two major comments follow and addressing them is recommended.

*Major Comments*.

1. Given the importance of the finding on the relative predictability of the blocked and zonal regimes, I would suggest giving a bit of history on it. Overall, the review on low-frequency variability (LFV) of the midlatitude atmosphere in the paper's introduction is quite careful and complete. But the authors might wish to emphasize the fact that Legras & Ghil (*JAS*, 1985) were the first to find the greater stability and hence predictability of zonal flows in their 25-variable barotropic model on the sphere. This result was followed in the experimental paper of Weeks et al. (*Science*, 1997), using a barotropic rotating annulus, by a study of the variability and persistence of the laboratory blocked flow that essentially confirmed the findings of Legras & Ghil (1985); see especially Fig. 5 in Weeks et al. (1997). So did the Lucarini & Gritsun (*Clim. Dyn.*, 2020) paper, which used the three-layer quasi-geostrophic model of Marshall & Molteni (*JAS*, 1993) and the methodology of unstable periodic orbits

(UPOs). The fact that Lucarini & Gritsun (2020) used a baroclinic model removes the doubts about the greater stability of zonal flows being exclusively due to the barotropic character of the Legras & Ghil (1985) model and of the Weeks et al. (1997) apparatus.

It would be of particular interest if the authors of the present paper could take a closer look at baroclinic vs. barotropic effects in their model, with respect to this question of the relative stability and persistence of blocked vs. zonal flows, when the two types of regimes coexist. See also the discussion in Ghil & Lucarini (*Rev. Mod. Phys*., 2020, p. 035002-36).

2. The authors refer to using a "machine learning algorithm called Gaussian Mixture Clustering (GMC)," which is described in Appendix A. While machine learning and AI are all the rage these days, I'd be curious to know how this algorithm differs from the one that was used on observational data by Smyth et al. (*JAS*, 1999).

Michael Ghil

---

## Author Response (AR2)

Dear Editor,

We hope this message finds you well.

We are writing to express our heartfelt gratitude for the acceptance of our manuscript with minor corrections. Your constructive feedback and insightful suggestions have been invaluable, and we are truly appreciative of your support throughout this process.

We have diligently incorporated all the suggestions you provided, ensuring that the manuscript now fully aligns with the high standards of Earth System Dynamics. We believe these adjustments have significantly enhanced the clarity and quality of our work.

Furthermore, we would like to extend our sincere thanks to the reviewers for their thorough and thoughtful critiques. Their valuable insights and constructive comments have greatly contributed to the improvement of our manuscript, making it a stronger and more robust piece of research.

We are attaching the revised manuscript, which includes all the recommended changes. We are confident that the manuscript now meets the expectations of the journal and hope it will be well-received by the readers.

Thank you once again for your guidance and for giving us the opportunity to refine our work. Should you need any further information or assistance, please do not hesitate to contact us.

Warm regards,

Anupama K. Xavier, Jonathan Demaeyer, Stéphane Vannitsem

Response to the Editor:

1. Regarding reviewer 1's comment on the complexity of interpreting Lyapunov exponents in spatio-temporal systems: While you addressed the reviewer's comment in your response, I would encourage you to also make a brief note in the manuscript of the fact that the LE here consider both spatial and temporal chaos. This will prevent other readers going into the manuscript with the idea that the LE here only signifies temporal chaos.

Thank you very much for pointing this out. Relevant explanation is added from line 190 to 195 in the manuscript in the model characteristics session as "Each basis function represents corresponding spatial patterns.Therefore this configuration yields a set of 30 variables, including 10 barotropic variables, 10 baroclinic variables, and 10 ground temperature variables. Note that as explained in the qgs documentation, the basis functions of the model can be easily altered. In the context of this model, the instability properties, i.e. the Lyapunov vectors, are affecting all the spatial modes at once, and are therefore characterising the spatio-temporal chaotic evolution of small perturbations. The approach adopted here is similar to the one used for instance in Vannitsem and Nicolis (1997)" To give a connection "As Mentioned in section 2.1, Lyapunov exponents computed here consider both spatial and temporal chaos." is added into line 276 which explains the theory behind the computation of Lyapunov exponents.

2. Figure B4: A significant percentage of people are colour-blind and cannot distinguish between red and green. I would therefore suggest re-running your plotting scripts with modified colours (e.g., choosing blue instead of green). This way, a greater number of people will be able to read and appreciate your work.

Thank you very much for this suggestion. We changed the colour palette of the figures suitable for people with colorblindness, especially figure B4.

[Figure]

3. Appendix A: I recommend also citing Smyth et al. 1999 (see reviewer 2's second comment) when describing the Gaussian Mixture Clustering method. It may help readers recognise a familiar method.

Thank you. Added the reference in the corresponding section

First of all, thank you very much for the helpful comments and suggestions you have made about our manuscript. This will definitely help the improvement of the research to a greater extent.

**Reviewer 1**

Major comments:

1. One major concern pertains to the classification of three weather regimes using a machine learning (ML) method and the correlation of each identified regime with stability or instability, determined by averaged local Lyapunov exponents. While Charney and Devore's (1979) model is traditionally used for studying non-chaotic weather regimes, Lorenz's (1963a) model is renowned for illustrating chaotic features. It's crucial to note that Lorenz applied a similar approach in the early 1960s to study chaotic and nonlinear oscillatory solutions in a two-layer quasi-geostrophic system (e.g., Lorenz 1962, 1963b; Shen, Pielke Sr., and Zeng, 2023, https://www.mdpi.com/2073-4433/14/8/1279 ). A brief review of Lorenz's contributions is necessary, and a comparison of the QG-based models should address the variation in the number of Fourier modes concerning chaotic features.

Thank you very much for the suggestion. It is indeed important to discuss Lorenz's contributions and more precisely the comparison between QG - based models in the present context. More elaborated discussion is added in the introduction section of the manuscript from line 45 to 75 as follows

"The pursuit of simplified models for atmospheric phenomena has a long history, dating back to Lorenz's seminal work in the early 1960s (Lorenz, 1960, 1962, 1963a, b). This approach recognizes the value of sacrificing some detail in exchange for a deeper grasp of fundamental physical processes. Lorenz demonstrated the power of this strategy by leveraging Fourier series to distill the barotropic vorticity equation into three ordinary differential equations (Lorenz, 1960). These equations, while omitting smaller scales of motion, yielded valuable insights into atmospheric scenarios such as flow interactions

and current stability. Subsequently, he developed a simplified geostrophic model using truncated Fourier-Bessel series (Lorenz, 1962). This eight-equation model captured baroclinic instability, a critical process in atmospheric dynamics, while maintaining key energy relationships. Notably, the model successfully reproduced observed flow regimes and transitions in rotating fluids, suggesting its effectiveness in studying large-scale atmospheric behavior. Lorenz's 1963 research yielded significant advancements in our understanding of atmospheric dynamics through two key publications (Lorenz, 1963a). The first introduced a now-iconic system of three differential equations, derived from a further simplified model for fluid flow. This groundbreaking work unveiled the concept of sensitive dependence on initial conditions, a cornerstone of chaos theory.

In the same year, Lorenz explored a separate avenue by investigating a simplified model for symmetrically heated rotating viscous fluids (Lorenz, 1963b). This work resulted in a system of fourteen ordinary differential equations governed by two external parameters: the thermal Rossby number and the Taylor number. Analytical solutions revealed the existence of purely zonal flow and superimposed steady waves, while numerical integration unveiled a richer tapestry of flow behaviors. Oscillatory solutions with periodic shape changes and irregular non-periodic flow emerged. Interestingly, increasing the Taylor number generally led to greater flow complexity, except at very high values where the model's truncations became unrealistic.

Perhaps most intriguing was the coexistence of unstable purely zonal, steady-wave, and oscillatory solutions. This suggests intricate flow dynamics, with transitions between symmetric and unsymmetric vacillation occurring independently of instability. These findings highlight the ability of simplified models to unveil complex and nuanced behaviors in atmospheric dynamics (Shen et al., 2023).

Lorenz's pioneering work in the early 1960s demonstrated the power of simplified models for understanding atmospheric dynamics. By strategically neglecting certain complexities, he was able to capture key phenomena like baroclinic instability and chaos. However, for large-scale atmospheric simulations, computational efficiency becomes paramount. This is where QG (quasi-geostrophic) models come in. QG models prioritize large-scale features by making specific approximations, allowing for rapid simulations and analyses of broad atmospheric circulation patterns. While they may not capture the intricate details explored by Lorenz's models, QG models remain a workhorse for studying large-scale atmospheric phenomena."

Several issues need attention:

1. Gaussian Mixture Clustering (GMC) was employed for classification, assuming each cluster has a Gaussian component. The suitability of this assumption for chaotic regimes should be addressed, especially considering the regular spatial patterns that appear in the classified regimes. Associating different weather regimes with components of the leading Lyapunov vector, particularly in the presence of multiple positive Lyapunov exponents, should be addressed. The challenge is heightened by the time-varying components of each Lyapunov vector along the solution orbit, making it difficult to link specific components to zonal or blocking regimes.

Thank you very much for pointing it out. The distribution of the data points on the attractor is not presumably a Gaussian distribution. But the algorithm itself approximates the distribution as Gaussian and identifies clusters based on that approximation. This is pointed out in the manuscript from line 442 to 445 as

"While the distribution of data points on the attractor may not align with a Gaussian distribution, the algorithm proceeds by approximating the distribution as Gaussian and subsequently identifies clusters based on this approximation."

The methodology depicted in this paper is such as computing the local Lyapunov exponents on each datapoint on the attractor, clustering the attractor using GMC and then calculating the average of largest local Lyapunov exponents based on the identified cluster. Even though the attractor has multiple positive Lyapunov exponents, we only investigated the first one since the dynamics of the error will ultimately follow this exponent. To clarify this we added in the manuscript

"Although the attractor exhibits multiple positive Lyapunov exponents, our investigation focused solely on the first exponent, as it governs the dynamics of the error and is therefore considered the most influential."

2. The use of a fixed cluster number (3) in GMC for models with different numbers of positive Lyapunov exponents (LEs) (e.g., 3 positive LEs in Figure 7 and 12

positive LEs in Figure 12) raises concerns. Various clustering values should be explored to illustrate the relationship between the number of weather regimes and the number of positive LEs. For example, can we observe similar weather regimes in the 30 and 165 variable systems? If this is the case, does it imply a consistent number of multiple regimes or equilibrium points across both systems?

Thank you for your input. Evaluating various cluster numbers in the clustering process is crucial for obtaining meaningful outcomes. Our experimentation, spanning from 2 to 6 clusters, revealed distinct flow regimes for 2 and 3 clusters, each exhibiting notable differences. However, as the cluster number increased to 4, we observed overlap between two clusters, resulting in identical structures and flow patterns. This trend persisted with further increases in cluster number. Consequently, we concluded that the attractor achieved optimal clustering with clear and nearly equal data point distribution when utilizing 3 clusters. This is added in the manuscript in the lines 225 to 235 as

"Through experimentation encompassing 2 to 6 clusters, discernible flow regimes emerged for 2 and 3 clusters, showcasing significant distinctions. However, as the cluster count reached 4, we noted convergence between two clusters, leading to identical structures and flow characteristics. This trend persisted with additional cluster increments. Hence, we inferred that the attractor attained optimal clustering with evident and nearly uniform data point distribution when employing 3 clusters."

3. Is it possible to compute Lyapunov exponents for individual weather regimes? Could the time evolution of each weather regime be graphed for comparative analysis?

Thank you very much for this suggestion. In our current analysis, we're computing local Lyapunov exponents for every data point on the attractor. These exponents help us understand the stability of trajectories in our system. Once we've classified the data points using Gaussian mixture clustering, we determine the average of the largest local Lyapunov exponent within each cluster. This process allows us to identify the predictability horizon associated with each cluster. Furthermore, we analyze the flow regimes by examining the average geopotential height at 500 hPa within each cluster. This helps us characterize the different atmospheric circulation patterns that emerge from the data. By combining these approaches, we gain insights into both the predictability and the underlying dynamics of the system, enabling a deeper understanding of its behavior.

4. Concerning predictability in typical dynamical systems, characterized by systems of ordinary differential equations (ODEs), a positive Lyapunov exponent (LE) usually signifies temporal chaos (distinct from spatial-temporal chaos). In estimating predictability horizons under different conditions, a higher positive LE, on average, implies a greater average growth rate, indicating faster error growth and thus diminished predictability horizons. However, when applying this concept to assess predictability in spatial-temporal systems, it becomes imperative to account for errors related to spatial movement. The analogy of whether more intense hurricanes (with higher growth rates) are less predictable encourages authors to contemplate the influence of spatial movement on error predictions. Therefore, in contrast to a zonal flow, although a blocking regime is linked to instability manifested by a larger LE, the consideration of spatial movement is essential when comparing errors in zonal and blocking cases in order to compare their predictability horizons.

The QGS land atmosphere coupled model is a spectral model where we project ODEs on several basis functions that represent different spatial patterns. Precisely in the manuscript we used 10 different basis functions which represent the spatial pattern shown in the figure above. So computing the perturbations of the spatial mode or computation of

[Figure]

**Figure 1.** *The first 10 basis functions $F_i$ evaluated on the nondimensional domain of the model*

The Lyapunov exponents in the spectral space are indeed considering both spatial and temporal chaos. In this context, spatial amplification of the modes will not be calculated. As you mentioned, the spatial evolution of error is of considerable interest, its significance becoming more pronounced in higher resolution runs, which we intend to

explore in a subsequent investigation. The figure shown above depicts the first 10 basis functions $F_i$ evaluated on the nondimensional domain of the model.

Specific Comments.

Page 5: Please indicate whether and how Eqs. (1) and (2) are coupled with Eqs. (3) and (4).

Thank you for the comment. Discussion about the coupling is included in the revised manuscript from line 177 to 185 as

"The hydrostatic relation in pressure coordinates $\partial \Phi = -1/\rho a$ where the geopotential height $\Phi_i = f_0 \psi^i_a$ and the ideal gas relation $p = \rho_a R T_a$ allow one to write the spatially dependent atmospheric temperature anomaly $\delta T_a = 2f_0\theta_a/R$ ,with $\theta_a$ as the baroclinic stream function. This can be used to eliminate the vertical velocity $\omega$. This changes the independent dynamical field to the stream function field $\psi_a$ and the spatially dependent temperatures $\delta T_a$ and $\delta T_g$."

Page 7: Please indicate the equation(s) that contains Cg.

Indicated in the revised manuscript as

"The dimensional meridional differential shortwave solar radiation absorbed by the land and the atmosphere are given by $\delta R_g = \sqrt{2}C_g\cos(y/L)$ and $\delta R_a = \sqrt{2}C_a\cos(y/L)$ respectively. Hence we decide to provide $C_a = 0.4C_g$. The variable $C_g$ is a dimensional parameter, which is an indicator of the meridional difference in solar heating absorbed by the land between the walls, and it is the crucial parameter in our land atmosphere coupled model."

Page 7, Figure 1c with ACC.  Is it possible to identify a time scale of approximately 36 days within the timeframe spanning 20 to 80 days?

[Figure]

*Figure 2. (a) Autocorrelation of barotropic ($\psi_{a,1}$) atmospheric streamfunction and the ground temperature ($T_{g,1}$) for C g = 300 Wm$^{-2}$and $k_d$= 0.085 (b) Powerspectrum of the same variables*

[Figure]

*in $log_{10}$ scale.*

*Figure 3. Power spectrum same as fig. 2 against time period in normal scale instead of $log_{10}$ scale*

The figure below depicts the autocorrelation and power spectrum of barotropic ($\psi_{a,1}$) atmospheric streamfunction and the ground temperature ($T_{g,1}$) for $C_g = 300$ Wm$^{-2}$ and $k_d = 0.085$. Power spectrum in the first panel has frequency and time period depicted in the respective x-axis in a $\log_{10}$ scale as in the second panel time period depicted in the normal scale in order to clearly identify the peaks.

Through a comprehensive examination of the power spectrum, we found it inconclusive to find oscillations every 36 days since we are observing several peaks during the course of 80 days.

Page 7: Can a figure analogous to Figure 1 be generated for each of the categorised regimes?

Thank you for the comment. Yes, it could be drawn by giving different colors to the data points belonging to different flow regimes or respective clusters. In the figure below, phase space dynamics of the model projected on the ($\psi_{a,3}$, $\psi_{a,2}$)-plane is shown in panel (a)

[Figure]

*Figure 4. The phase space dynamics of the model projected on the ($\psi_{a,3}$, $\psi_{a,2}$)-plane is shown in panel (a) for $k_d = 0.08$. The figure illustrates the Gaussian mixture clustering results, where each data point is colored according to its corresponding cluster or flow regime. Panels (b), (c), and (d) depict the temporal evolution of barotropic ($\psi_{a,1}$) and baroclinic ($\theta_{a,1}$) atmospheric stream function along with the ground temperature ($T_{g,1}$) for $C_g = 300\ Wm^{-2}$ and $k_d = 0.08$ with colors corresponding to their respective clusters.*

for $k_d = 0.08$ . The figure illustrates the Gaussian mixture clustering results, where each data point is colored according to its corresponding cluster or flow regime. Panel (b), (c) and (d) represents the temporal evolution of barotropic ($\psi_{a,1}$) and baroclinic ($\theta_{a,1}$) atmospheric stream function along with the ground temperature ($T_{g,1}$) for $C_g = 300\ Wm^{-2}$ and $k_d = 0.08$ with colors corresponding to their respective clusters. We included this figure into the appendix as another way of representing clusters.

Page 9: The discussions on the Oseledec method should be expanded to incorporate insights from Lorenz's contributions. (e.g., Lorenz 1965; please see a review by Shen, Pielke Sr, and Zeng, 2023).

Thank you very much for drawing our attention toward this paper. In Lorenz's 1965 paper, he uses a 28-variable atmospheric model which is developed by extending the equations of a two-level geostrophic model using truncated double-Fourier series. This model accounts for nonlinear interactions among disturbances of varying wavelengths. Numerical integration is employed to find nonperiodic time-dependent solutions. By comparing solutions with slightly different initial conditions, the rate of growth of small initial errors is investigated. Lorenz's error growth estimation is based on using singular value decomposition which is not the direction we wanted to proceed with the current paper as we computed the Lyapunov exponents that are asymptotic properties of the attractor. Hence we will not use that reference in the current context.

Page 11: Please add Figure B3 to include flows for Cg = 400 or kd = 0.12, for periodic flows.

[Figure]

**Figure 5.** *Phase space dynamics of the model projected on the ($\psi_{a,3}$, $\psi_{a,2}$) - plane is shown in panel (a). Gaussian mixture clusters covariance are represented with orange, green and red ellipsis, for $k_d$= 0.105, the system exhibits periodic behavior that results in non-identifiable flow regimes and indistinguishable clusters.*

System enters into periodic behavior when $k_d$ = 0.105 to 0.115. Hence we added figure B3 with $C_g$=300 W/m$^2$ and $k_d$= 0.105 which depicts Phase space dynamics of the model projected on the ($\psi_{a,3}$, $\psi_{a,2}$) - plane shown in panel (a). Gaussian mixture clusters covariance are represented with orange, green and red ellipsis. but for $k_d$= 0.105, the system exhibits periodic behavior results in to non identifiable flow regimes and indistinguishable clusters

Pages 12 and 13: while $\lambda_1$ and $\lambda_2$ are used for representing the 1st and 2nd LEs, respectively, the symbol lambda indicates heat exchange. Please consider making changes to reduce confusion.

Thank you very much for pointing this out. $\lambda_1$ and $\lambda_2$ are changed into $LE_1$ and $LE_2$ for

[Figure]

***Figure 6.*** *First and second Lyapunov exponents of the land atmospheric coupled model for*

*different values of $C_g$ and $k_d$. The blue line represents the first and the orange line represents the second Lyapunov exponent respectively. The values of the Lyapunov exponents are given in $days^{-1}$*

avoiding confusions in figure 6 and 8 which actually portraits

Page 12, line 280. Does the selection of lambda = 0 result in an uncoupled model? How can this be contrasted with the models proposed by Charney and Devore, Lorenz (1962) and/or Lorenz (1963b)?

Selection of lambda = 0 will not result in an uncoupled model. It only ceases the heat exchange between land and the atmosphere. The land and atmosphere components in the model are still interacting via incoming shortwave radiations, outgoing long wave radiations as per the equations. The model proposed by Charney and Devore has similar dynamics where the only difference is the energy balance system. The land atmosphere coupled model has a realistic energy balance system as in Barsugli and Battisti (1998). The coupling of the atmospheric components with the ground is constituted by the surface friction and the radiative flux whereas Charney and Devore are using an energy balancing system based on Newtonian cooling coefficient.

Pages 12 & 17, (in Figures 7 & 12), please offer perspectives on whether the presence of the plateau suggests the existence of singular eigenvalues with higher multiplicity.

[Figure]

***Figure 7.*** *Lyapunov spectrum and standard deviation of the Lyapunov exponents calculated using a bootstrap method. Lines with values +/- 2\*standard deviation (2\*std in the panels) of the Lyapunov exponent values is also depicted in the figure in the scale of the Lyapunov spectrum. The second panel is a zoomed version of the first figure only concentrating on the Lyapunov exponents that caused the plateau which indexed from 3 to 14*

The figures above depict the Lyapunov spectrum and standard deviation of the Lyapunov exponents calculated using a bootstrap method. Lines with values +/- 2\*standard deviation (2\*std in the figures) of the Lyapunov exponent values is also depicted in the figure in the scale of the Lyapunov spectrum. The second figure is a zoomed version of the first figure only concentrating on the Lyapunov exponents that caused the plateau which indexed from 3 to 14. The exponents do not show a large uncertainty band, and therefore one can now say confidently that the values forming the plateau are clearly distinct to each other, suggesting the absence of a potential degeneracy of the eigenvalues.

**Reviewer 2**

Given the importance of the finding on the relative predictability of the blocked and zonal regimes, I would suggest giving a bit of history on it. Overall, the review on low-frequency variability (LFV) of the midlatitude atmosphere in the paper's introduction is quite careful and complete. But the authors might wish to emphasize the fact that Legras & Ghil (JAS, 1985) were the first to find the greater stability and hence predictability of zonal flows in their 25-variable barotropic model on the sphere. This result was followed in the experimental paper of Weeks et al. (Science, 1997), using a barotropic rotating annulus, by a study of the variability and persistence of the laboratory blocked flow that essentially confirmed the findings of Legras & Ghil (1985); see especially Fig. 5 in Weeks et al. (1997). So did the Lucarini & Gritsun ( Dyn., 2020) paper, which used the three-layer quasi-geostrophic model of Marshall & Molteni (JAS, 1993) and the methodology of unstable periodic orbits (UPOs). The fact that Lucarini & Gritsun (2020) used a baroclinic model removes the doubts about the greater stability of zonal flows being exclusively due to the barotropic character of the Legras & Ghil (1985) model and of the Weeks et al. (1997) apparatus.

Thank you very much for the suggestion. As per the comment we included a more elaborated introduction including all the mentioned literatures in the revised version of the manuscript from lines 105 to 120. The included portion is

"Legras & Ghil (JAS, 1985) employed a higher-order barotropic spectral spherical model to investigate blocking and zonal flow regimes dynamics, suggesting that their model displayed properties akin to an index cycle, and later stochastic forcing was introduced to Charney's deterministic model, leading to transitions between high- and low-index states (Benzi et al., 1984; Egger, 1981; Sura, 2002). The impact of stochastic forcing on the stability of atmospheric regimes was also recently considered in a highly-truncated barotropic model by Dorrington and Palmer (2023), where they provide a mechanism to explain the increased persistence of blocking due to the noise in such simple models.

In this paper, Legras and Ghil (1985) also discussed the realistic existence of blocked and zonal flow regimes which are obtained as unstable stationary solutions due to the barotropic influence of the LFVs in the atmosphere. More persistent zonal flows are also identified in several occasions which seems to be a deviation from the earlier studies.

Later the stability studies by Weeks et al. (1997) recreating zonal and blocked regimes in an experimental annulus setup further substantiated the findings of Legras and Ghil (1985).

Schubert and Lucarini (2016)'s numerical investigation employing a QG model revealed a counter-intuitive finding that during blocking events, the global growth rates of the fastest growing covariant Lyapunov vectors (CLVs) are significantly higher, indicating stronger instability compared to typical zonal conditions. The difficulty in predicting the specific timing of blocking onset and decay further contributes to the observed instability behavior, aligning with Kwasniok (2019) findings associating anomalously high values of finite time largest Lyapunov exponents with blocked atmospheric flows.

Lucarini and Gritsun (2020) demonstrated that blocking phenomena exhibit higher instability compared to typical atmospheric conditions, irrespective of whether they occur in the Atlantic, Pacific, or globally. This analysis utilized the simplified atmospheric model proposed by Marshall and Molteni (1993) and assessed stability based on unstable periodic orbits (UPOs). Importantly, this research dispelled the misconception that the increased stability of zonal flows solely resulted from the barotropic nature of the model in the study of Legras and Ghil (1985) and Weeks et al. (1997) apparatus. Consistent results were obtained by Faranda et al. (2016, 2017), utilizing extreme value theory for dynamical systems, which identified blocking regimes with unstable fixed points in a heavily reduced phase space. Their findings indicated that blockings exhibit higher instability in the circulation, linked to an increased effective dimensionality of the system. This agreement with Lucarini and Gritsun (2020) study further supports the notion that blocking events display stronger turbulence and instability, challenging conventional expectations."

It would be of particular interest if the authors of the present paper could take a closer look at baroclinic vs. barotropic effects in their model, with respect to this question of the relative stability and persistence of blocked vs. zonal flows, when the two types of regimes coexist. See also the discussion in Ghil & Lucarini (Rev. Mod. Phys., 2020, p. 035002-36).

This is an interesting comment. Indeed it is really important to analyze the stability of flow regimes that were influenced by the barotropic or baroclinic part of the model. In the discussion of Ghill & Lucarini (2020), they were pointing out 4 different possibilities to obtain an unstable blocking event compared to the zonal flow which includes slowing down of Rossby waves or their linear interference, the existence of multiple flow equilibria resulting slower flow regimes, the idea of oscillatory instabilities of one or

more of the multiple fixed points that can play the role of regime centroids and the last one was the formation of blocking events when the trajectory is near an extremely unstable periodic orbits (UPOs). As this comment says, instability could also be because of the barotropic or baroclinic part of the model. Inorder to find out that, we tried to average the barotropic and baroclinic stream function with respect to each cluster to identify the existence of different stream function values between zonal and blocking representing clusters and also studied the distribution of each baroclinic and barotropic mode But the results are inconclusive which denotes that it needs a more extensive methodology which we will pursue in our future works.

The authors refer to using a "machine learning algorithm called Gaussian Mixture Clustering (GMC)," which is described in Appendix A. While machine learning and AI are all the rage these days, I'd be curious to know how this algorithm differs from the one that was used on observational data by Smyth et al. (JAS, 1999).

We were really sorry to use the term 'Machine learning' since it is more of a data driven algorithm which we were corrected in the revised manuscript. Apparently the idea is the same as that in the paper Smyth et al., 1999.